# A Hardware Encoder-Based Synchronization Method for a Fast Terahertz TDS Imaging System Based on the ECOPS Scheme

**DOI:** 10.3390/s24061806

**Published:** 2024-03-11

**Authors:** Marcin Maciejewski, Kamil Kamiński, Norbert Pałka

**Affiliations:** Institute of Optoelectronics, Military University of Technology, 2 Kaliski Street, 00-908 Warsaw, Poland; kamil.kaminski@wat.edu.pl (K.K.); norbert.palka@wat.edu.pl (N.P.)

**Keywords:** terahertz imaging, synchronization, time-domain spectroscopy, nondestructive testing

## Abstract

In this paper, we report our use of a hardware encoder-based synchronization method for a fast terahertz time-domain spectroscopy raster scanner built with the commercially available TeraFlash Smart platform. We describe the principles of our method, including our incorporation of synchronization signals from various devices included in the scanner. We also describe its implementation in a microcontroller with a dedicated counter. By such means, a fast scanning mode was obtained, which was 35 times faster than a traditional step-by-step approach. To validate the proposed synchronization method, we carried out measurements using the USAF 1951 resolution test and a fiberglass plate with a set of intentionally introduced defects. Our results confirmed that the TDS scanner with the developed synchronization method was able to capture high-quality images with resolutions as high as those obtained using traditional step-by-step scanning, but with significantly reduced scanning times.

## 1. Introduction

Quality control is important in all stages of production processes. Because producers seek to manufacture better products while also minimizing costs, the use of nondestructive testing (NDT) has become an increasingly important testing method. There are many different techniques of nondestructive testing [1,2,3,4,5]. Each of them has distinctive features, so that one single technique may be best suited for measuring a given group of materials or parameters. In the case of modern fiber-reinforced plastics and foams, as well as sandwich and hollow structures, THz-based methods seem to be a reasonable choice for fault detection and structural analysis [6]. In contrast to X-ray radiation, THz radiation is safe for humans [7], ensures a higher spatial resolution than microwaves [8], involves less Mie and Rayleigh scattering than infrared, and allows for non-contact measurement, unlike ultrasound [9,10]. The above advantages mean that methods based on THz waves can provide very good three-dimensional images of materials, which reveal details such as individual layers, glued joints between layers, and cavities [8,11,12].

Several methods are used for imaging in the THz band [13,14,15], the most popular of which are frequency-modulated continuous waves (FMCW) [16], cameras [17], Terahertz inverse synthetic aperture radars [18], and time-domain spectroscopy (TDS) [19]. The last of these methods is especially widely used for NDT, due to the very good spatial and depth resolution which it provides [6,20,21]. In recent years, this method has become even more important as a result of technological developments [22]. Until recently, the major problem associated with the TDS method was the need to use a mechanical delay line in the measurement system. The frequency of its tuning and the resulting data rate, usually in the order of several Hz, presented a serious obstacle to the implementation of TDS in fast imaging systems. The solution appeared with the development of synchronous laser-based systems, including TDS based on electronically controlled optical sampling (ECOPS) [23]. Such systems use two femtosecond lasers, one of which is modulated by a piezoelectric element. This enables the achievement of a significantly higher data rate (in the order of kHz), and thus significantly speeds up the imaging process [24].

The TDS system provides point data; imaging therefore requires a measuring system with a mechanical scanner which moves the head over a sample. To obtain maximum benefit from an ECOPS system, its high data rate should be translated into a high pixel-acquisition rate. To achieve this goal, the mechanical system should operate at a high speed, and the acquisition system must enable the assignment of 2D coordinates to the given result, indicating the place of its capture. The easiest way to achieve this is by triggering the measurement in precisely defined positions of a Cartesian scanner during line-by-line scanning movement. Unfortunately, TDS ECOPS systems work with a fixed sampling rate after starting up; as a consequence, they cannot be triggered. Therefore, another means of synchronizing the movement of the scanner with THz data acquisition is necessary.

One possibility is to use a spiral scanning trajectory (Figure 1a) instead of a Cartesian trajectory. This allows for fast movement, and also considerably simplifies the synchronization problem, by eliminating accelerated motion from the scanner model. Practically, this system can be implemented by placing a sample on a rotating base and moving the measuring head along its radius [25,26]. The speeds of both elements during the scan of the entire sample are homogenous, thus eliminating any need for an additional synchronization method. However, this solution seems to be problematic in the case of larger samples, because a constant rotational speed results in a significant difference in pixel density between the center and the edge of the sample. In addition, there may be an imbalance in the rotation system in the case of a heavy sample with an uneven distribution of mass.

A raster scan performed according to the Cartesian scheme (Figure 1b) eliminates the uneven distribution of samples and the problem of imbalance associated with a rotary setup. Moreover, it allows any selected part of the sample to be used. A Cartesian scan, however, implies nonuniform motion, and this generates two further problems. Firstly, this movement pattern is not optimal because it requires acceleration and deceleration of the sample or the head (depending on which element is movable), so that some fragments are scanned at a lower speed, extending the scanning time. However, by using high-thrust motors, it is possible to obtain high linear acceleration values of the scanner axis, and significantly reduce the impact of motion heterogeneity on the scanning time. The second problem is the need for synchronization. Nonhomogeneous speeds and short stops of motors of indeterminate duration prior to changes in scanning direction mean that, in the absence of an additional mechanism, it is impossible to assign a given result to the measurement point on the sample.

The synchronization problem has been solved by using a marker, in the form of a metal ruler located at the edge of the scanning area [27]. This gives a characteristic result, the presence of which in the data series of each image line enables alignment of individual lines in the final image of the scanned sample. This solution, however, requires that each line contains a signal reflected from the marker, so it does not allow scanning of a freely chosen fragment of the sample. It also requires adjustment of the marker height to a given sample, and additional signal processing, leading to longer scanning times. Therefore, to take full advantage of the scanner in the Cartesian scheme, it is necessary to implement hardware synchronization of the mechanical elements of the scanner with the THz data acquisition system.

For the present study, we developed a hardware encoder-based synchronization method for a THz raster scanner, which operates according to the Cartesian trajectory scheme. The proposed method assumes the use of a microcontroller (MCU) to read the output states typically available in this type of system, i.e., encoder signals from the scanner axis controller and triggers signaling the next THz pulse, which are often available in TDS systems. Based on the read states, the device generates information that enables linkage of a given result with the measurement point on the sample. In the present study, a synchronization system built in accordance with the above assumptions was successfully implemented in a TDS Cartesian scanner. The obtained results confirmed the possibility of scanning samples with dimensions of 400 × 450 mm, with a step size of 1 mm, in about 37 min.

The paper is structured as follows: Section 2 describes the Cartesian TDS scanner used for data acquisition. Section 3 describes the principles of the proposed synchronization method and its implementation. Measurement results are presented in Section 4, followed by a conclusion in Section 5.

## 2. Cartesian THz Scanning Setup

The measurement setup considered in this work consisted of a main controller (PC computer), a mechanical system comprising an XY scanner (gantry) and a motor controller, a TDS TeraFlash Smart system provided by Toptica Photonics from Munich, Germany, and a synchronization device (Figure 2a). The source of THz data in the system was the TeraFlash Smart platform, a TDS device which was operated according to the ECOPS scheme. The system used two 25 mW femtosecond lasers, operating synchronously. One of the lasers was modulated at 1600 Hz, which corresponded to the acquisition frequency *f_r_* of the THz pulses. Each pulse consisted of 3125 samples and lasted for 154 ps. The pulse reflected from a flat reference mirror is shown in Figure 2c, in which an amplitude of about 365 units and a half-width of 0.35 ps can clearly be seen. Averaging with jitter correction was used to reduce random noise components in the signal.

The considered setup was equipped with a reflection head (Figure 2b), which was connected to the main unit by polarization-maintaining optical fibers. In the head, four off-axis parabolic mirrors were used, directing the beam towards the sample at an angle of 8 degrees. The size of the beam spot at the focal distance was about 1 mm. The head was purged with dry air to remove any unwanted influences from water vapor.

For scanning flat samples, a gantry system was constructed, consisting of three linear motors (model LRT0750DL-E08CT3A manufactured by Zaber company from Vancouver, BC, Canada), each characterized by a maximum speed of 700 mm/s, a high maximum thrust of 2400 N, and unidirectional accuracy at a level of 0.2 mm. The tested samples were placed on an aluminum breadboard with dimensions of 600 × 600 mm; this was, in turn, mounted on two linear drives, thus forming the *Y*-axis of the scanner. The use of two motors working in parallel increased the rigidity of the breadboard and, additionally, enhanced the cumulative thrust of the axes, enabling a reduction in the time required for axis acceleration and deceleration. Above the examined sample, there was a head assembly on the third motor, set perpendicular to the previous two, thus forming the *X*-axis of the scanner. Between the head and the motor carriage, there was a linear table for changing the height of the head. Thanks to this, regardless of the thickness of the sample, the heads could be set to the distance of the maximum signal, so that the reflecting surface was at the focal distance of the head’s mirrors. A photo of the XY scanner is presented in Figure 3a.

The scanning process may be briefly described. First, the *X*-axis drive, called the slow axis, moves the head to the initial position. Then, the sample is moved along the *Y*-axis, called the fast axis, and data for the first line of the image are recorded. After that, the *X*-axis drive moves the head by the given step, and the next line is scanned. This procedure is repeated until the scan is completed. The scanning range for the *X*- and *Y*-axis can be freely changed from 0 to 600 mm. The minimum theoretical size of a scan step for the *X*-axis is a consequence of the micro step of the linear drive. In the present study, this was about 2 µm; however, due to the inaccuracy of the motor, and the size of the THz beam spot, the minimum step of the axis was set to 0.1 mm. For the *Y*-axis, the minimum step (Δ*y*) is a consequence of the set speed *v* of the *Y*-axis and the number of averaging *N*, as expressed by the following equation:(1)$$\Delta y=\frac{N\cdot v}{f_{r}}$$

The data from the TeraFlash device are then read through an intermediate application called the server. This application receives successively recorded THz pulses and buffers them in the computer’s memory. The scanner control program connects to the server and retrieves the pulse data in the order in which they were measured. However, because the data have no timestamps, it is impossible to directly assign them to a moment of time and thus to a fragment of the sample. Therefore, an additional synchronization system was developed.

## 3. Synchronization Method

Proper synchronization should be carried out mainly at the hardware level, to ensure that delays are small and predictable. In the present study, we started by determining the available synchronization inputs and outputs of individual devices included in the scanner system. The TeraFlash system driver and the stepper motor driver were analyzed.

The TeraFlash smart platform is equipped with trigger input (TRIG) and synchronization output (SYNC). Synchronization output generates a square-wave signal with a frequency equal to half of the set data-rate frequency (800 Hz in the described system). Each edge of this signal indicates the acquisition of the next THz pulse. In the present study, we faced the problem that the output signal appeared as soon as the modulation was turned on (preparation stage), regardless of whether the data were read by the server application or not. As a result, after initiating the data readout, it was impossible to determine with which edge of the SYNC signal the first THz pulse—and, thus, all subsequent pulses—was associated. The solution to the above problem was to use the TRIG synchronization input to trigger the transfer of THz pulse data to the server. Data readout was configured to initiate transmission after a specific edge appeared on the TRIG input. In this way, the SYNC edge corresponding to the first THz pulse was indicated as the first SYNC edge after the TRIG edge. The SYNC signals oscillogram is shown in blue in Figure 4.

The Zaber X-MCC drive controller is equipped with four configurable DO_X digital outputs. Two of these were used for synchronization in the scanner system. The first one (DO_1) was configured to change the output state to the opposite, after successive specified numbers of steps of the fast-axis motor (Y). This number of steps was dependent on the scanner step value (pixel size) and was defined as the quotient of the scanner step and the stepper motor minimal step, which for the LRT0750DL-E08CT3A drive is approximately 2 μm. An example of the waveform for this channel during drive deceleration is shown in orange in Figure 4. Based on this signal, impulses were assigned to parts of the image lines equidistant in space. The second output (DO_2) was configured to change state when the fast-axis drive reached the desired position. This output was used to determine the beginning and end of each scan line and allowed individual image lines to be separated from each other. Figure 4 shows the synchronization waveforms before the end of the line and the drive stopping. The DO_2 signal observed at that time would have a constant value and therefore was not included in the graph.

The proposed synchronization method involved counting all SYNC pulses and marking those which occurred immediately before the event signaled by the stepper motor controller (shift by a given step (DO_1) or end of line (DO_2)). In turn, the main system controller software counted and numbered the subsequently read THz pulses. If the reading was initiated with a TRIG signal, the number of the synchronization pulse coincided with the number of the THz pulse. Using information about the numbers assigned to characteristic pulses, the main controller selected them from all read pulses, and used them to create subsequent image lines. (Figure 5).

The above-described method was implemented using an STM32F303K8T6 microcontroller on the Nucleo F303K8 evaluation board developed by Stmicroelectronics from Geneva, Switzerland. This consisted of a power supply system, a programmer, and a USB–UART converter, which greatly simplified the design of the device. On the printed circuit board dedicated to the device, there were only SMA connectors for the TeraFlash controller and a terminal block connector with an output converter from X-MCC to the TTL standard. This converter was implemented based on a set of 74LS04 inverters, in accordance with the documentation for the X-MCC controller. A photo of the assembled system is shown in Figure 6b.

The most important MCU module was the TIM2 counter, the only 32-bit counter in the group of so-called general-purpose counters. The clock signal (CLK) for the counter was obtained from the SYNC output of the TeraFlash device and was configured so that both signal edges were active. As a result, the counter was incremented with each new THz pulse. The 32-bit length meant that the counter would only overflow after 30 days or more, a period much longer than the planned time of several hours for a single measurement. One of the MCU outputs (START) was connected to the TRIG input of the TeraFlash device and configured to generate a short combined pulse while also resetting the TIM2 counter. This output was used to trigger data reading.

The counter had three inputs configured in Input Capture mode; these were marked as IC_X. For each of them, both signal edges were active. These inputs caused the counter value to be written to local registers (IC_X_Reg) when an active edge occurred. Data from the registers were then written to separate IC_X FI-FO buffers in the microcontroller memory via the direct memory access (DMA) controller. The DO_1 and DO_2 outputs from the X-MCC controller were connected to the IC_1 and IC_2 inputs of the MCU, respectively. Thanks to this, the IC_1 FIFO buffer stored pulse numbers related to the shift in the fast axis (Y) by a given value, and the IC_2 FIFO buffer stored pulse numbers related to the beginning and end of the image line. The third input was reserved for future use in development of the software and the scanner itself. Data from the buffers were read in the main system controller by the virtual serial port (VCP) via the USB interface. A block diagram of the microcontroller configuration is shown in Figure 6a.

Data from the synchronization system and the TeraFlash device were read in dedicated software prepared in the 64-bit version of the LabView 21.0.1 environment. This software was used for controlling the movement of the X and Y drives, for assembling the acquired data into the resulting image, and for archiving these data. The software was also used to modify many parameters of the scanner, including the scanning range, step length, movement speed, and the number of averaged THz waveforms. The scanner can operate in the described synchronous mode and in the regular step-by-step mode, in which scanning takes place with the head stopping at a given point for the duration of data reading.

## 4. Experimental Results

To evaluate the synchronization method described above, the acquired experimental results were compared with results obtained using traditional step-by-step scanning. The tested sample was a 3″ × 3″, negative, USAF 1951 target from Edmund Optics (Figure 3b). This is made in the form of a float glass plate of 1.5 mm thickness with a vacuum-disposed layer of durable chromium. It contains elements from number 1 (from group −2) to number 3 (from group 9). The reflective layer has a scratch and dig surface quality of 20–10.

A scan of the USAF 1951 target was performed with a step of 0.5 × 0.5 mm. For a synchronous scan, the maximum movement speed of the *Y*-axis was 700 mm/s, and the maximum acceleration was 5000 mm/s^2^, being limited by the inertia of the breadboard and the maximum possible engine thrust. The movement parameters of the *X*-axis, which do not have such a critical impact on the measurement speed, were set to lower values: 200 mm/s for speed and 1000 mm/s^2^ for acceleration. This limited the forces acting on the THz head mounted on the drive of this axis. With regard to step movement, the same velocity and acceleration values were selected for the *X*- and *Y*-axis, i.e., 200 mm/s and 1000 mm/s^2^, respectively. Measurements were performed without signal averaging. Figure 7 presents C-scans in peak-to-peak mode.

The images in Figure 7 are highly similar. The main difference between them is the presence of some jagged horizontal lines in Figure 7b. The step-by-step scanning variant ensures that the pulse comes from a precisely given point in the sample. Hence, equal image lines may be seen in both directions. In the synchronous variant, samples were taken with constant distances of 0.437 mm for the selected parameters, in accordance with Equation (1). The collected results were then assigned to a grid of points with a mesh size of 0.5 by 0.5 mm. As a result, the pixels making up the horizontal lines in the image do not follow the same line in the sample. However, these differences are so small that they do not make it difficult to recognize details in the image.

Both images are characterized by similar dynamic ranges, ranging from 183 to 378 for image a and from 185 to 382 for image b; these may be translated into contrasts of 194 and 197, respectively. To facilitate the resolution assessment, a vertical cross-section was determined through the x = 59 mm lines containing all horizontal lines from the −1 group of the USAF 1951 test, and the horizontal lines from element 1 of group −2 (Figure 8). Horizontal lines were used because the fast movement concerned the *Y*-axis, and so we might here expect differences between synchronous and step-by-step scans. As a criterion for distinguishing between lines in a given group, we applied a decrease in local dynamics (the difference between light and dark lines) to 10% of maximum dynamics. The images obtained by both methods were found to have the same resolution; on both, the smallest recognizable element was element 2 from the −1 group, characterized by 0.561 lines per mm. The step-by-step scanning time was 2 h. The described synchronization method reduced this time to 3.5 min.

Such a significant difference in measurement time offers the possibility of devoting some scanning speed to improving the quality of the acquired image. To this end, two methods may be considered. One of them involves averaging of the recorded impulses; the other involves increasing the spatial resolution of the scan. The results of averaging from 10 results, and of changing the scanning step to 0.1 mm, are shown in Figure 8. It can be seen that, with this type of sample, averaging does not improve line data; however, a reduction in step does lead to improvement. This is confirmed by the fact that, in this scan, element 3 from the −1 group is the smallest recognizable element. Reducing the step requires a decrease in the speed of drive movement and an increase in the measurement time. For the described sample, the measurement with a 0.5 × 0.1 mm step lasted 10 min.

To evaluate the operation of the scanner based on the developed synchronization method using a real sample, a scan was performed with a specially prepared test sample. This was a 400 × 450 mm board made of fiberglass composite. The thickness of the sample ranged from 2.97 mm at the corners to 3.77 in the middle of the edge. A total of 25 flat-bottomed holes of diverse sizes and depths were made in the plate by milling. These holes served as structural defects that should be detected during scanning. The scan was performed from the reverse side of the plate. The locations of the holes, their sizes, and their depths are shown in Figure 9.

The sample was scanned using the developed synchronization method. The drive movement parameters were set as in the USAF 1951 test scan. The scan covered the entire plate with a small 5 mm intentionally added margin, i.e., a range from −5 to 460 mm in the *X*-axis, and from −5 to 410 mm in the *Y*-axis. The ranges were given assuming that, according to Figure 9, the point (X = 0, Y = 0) was located in the lower left corner of the sample. Margins were added to observe the sample edge. For both axes, the scanning step was 1 mm. The sample scanning time was 37 min. For comparison, a step-by-step scan was performed which lasted over 20 h.

Based on the obtained results, a series of C-scans was determined in which the pixels corresponded to the peak-to-peak values of the THz pulse at time intervals of 4.5 ps. The positions of subsequent intervals were set to the values of optical delays corresponding to the depths of subsequent defects (Figure 10a–e). Displaying the results for defects at different depths in one common C-scan was impossible, due to significant differences in the intensity of pixels resulting from greater attenuation for deeper defects. Below, Figure 10f shows a B-scan for a horizontal line passing through a medium-sized defect group (*Y*-axis of about 172 mm). All five defects are visible in the scan, with those placed deeper being less clear. In C-scans, image detection of spherical defects was carried out using the circle Hough transform (CHT) method. All 25 defects were detected correctly.

The parameters of detected defects were analyzed, and the effects were determined, as shown in Table 1. The results for various depths were averaged, giving medium parameters for groups of defects of a given size. Standard deviations were adopted as an estimation error. Data on the locations of defects, medium distances, and average locations of given groups of defects on the *Y*-axis coincided with the data from the sample specification. The medium diameters of individual defects determined from the image were smaller than those of actual defects by 1 to 3 mm. This result is due to the size of the THz beam (Figure 2b). A defect was clearly visible only when a significant part of the 1 mm wide beam fell on the hole. Only then, at a given depth (given delay), is there a clear reflection visible in the image as pixels with greater intensity. This means that, in the image, the edge of the defect shifts towards the center with a distance equal to the width of THz beam; as a result, the calculated defect diameters are slightly smaller than in reality.

## 5. Conclusions

In this paper, we propose an encoder-based synchronization method for a fast THz raster scanner, which works in accordance with a Cartesian trajectory scheme. We give a detailed description of a mechanical scanner with fast linear drives equipped with synchronization outputs, as well as a THz TDS measuring system, which was operated according to the ECOPS scheme at 1600 pulses per second. The principles of the proposed synchronization method based on hardware signals available in the measuring system are discussed. Its implementation in the microcontroller system, and in dedicated LabView software, is also considered. To the best of our knowledge, a similar synchronization method for the ECOPS THz TDS raster scanner has never been described before. To evaluate the scanner, which was operated in accordance with the developed synchronization method, comparative tests were carried out using a scanner which was operated according to the traditional step-by-step method. The results obtained for the USAF 1951 resolution test confirmed that the fast scan did not significantly differ from its more slowly obtained counterpart. The results obtained by both methods had the same resolution, and individual lines were not shifted, thus confirming the correctness of the assumptions underpinning the method and also the implementation of the method. To further confirm the correctness of our work, a real and large sample with crafted defects was also scanned. The image obtained as a result of scanning was clear, and the captured defects were also visible to the automatic image-pattern recognition algorithms.

We may state, then, that the developed synchronization method described in this paper allows for a scanning time which at maximum speed and acceleration is 30 times faster than that achieved with the traditional step-by-step method, for both small samples (scan time—3.5 min vs. 2 h) and large samples (scan time—37 min vs. 20 h). The images obtained are of decent quality, further confirming the possibility of using the ECOPS TDS system with the proposed synchronization device. The method proposed here is cheap and easy to implement on any system equipped with an appropriate counter. It can also be adapted for use in non-Cartesian scanning, for example, with cylindrical samples. Finally, because of its significantly faster scanning time, the method proposed here also allows for averaging, or reducing the scanning step, to further improve the quality of obtained images, while maintaining fast acquisition times.

The scanner with the proposed synchronization method can quickly examine samples of different sizes and weights. It can also effectively scan only selected sections of the sample with a selected resolution. Therefore, we can consider an imaging variant in which the entire sample is first roughly scanned at high speed and with a large step. Then, fragments suspected of having defects are examined more thoroughly. This scenario is not possible in spiral and marker-based synchronized scanners.

Further improvements to the method’s implementation are possible. In the current version, the software waits for the processing of data from the scanned line to finish before it starts another line scanning. Data reading and processing operations are parallelized, which reduces the waiting time, but still causes an unnecessary increase in overall scanning time. In subsequent versions, further parallelization of measurement and data analysis operations is planned, so that there will be no need to stop the scanner to finish processing the data of a given line. This should result in further reductions in scanning time.

## Figures and Tables

**Figure 1 sensors-24-01806-f001:**
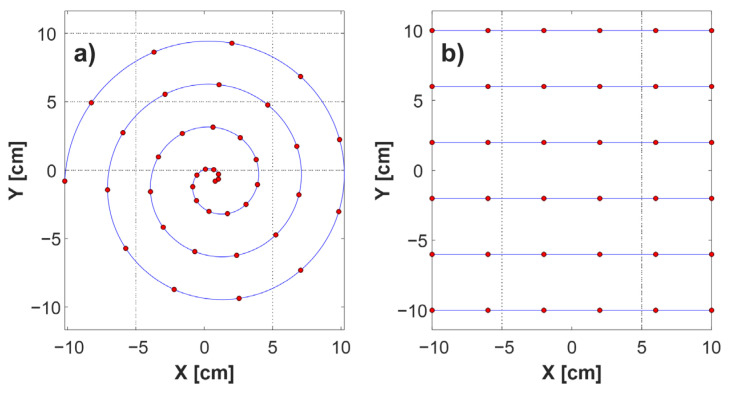
Comparison of scanning trajectories. (**a**) Spiral; (**b**) Cartesian. Blue lines indicate the scanner’s movement paths, and red dots symbolize data acquisition points.

**Figure 2 sensors-24-01806-f002:**
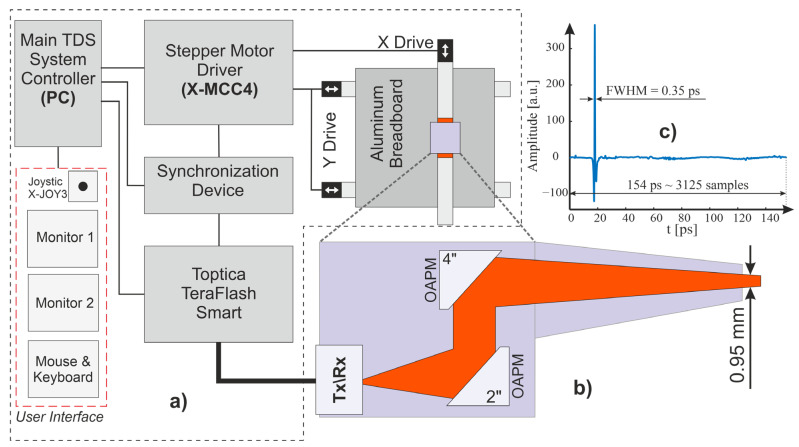
(**a**) Block diagram of the THz scanner system; (**b**) diagram of the reflection head; (**c**) THz signal reflected from the gold-plated reference mirror.

**Figure 3 sensors-24-01806-f003:**
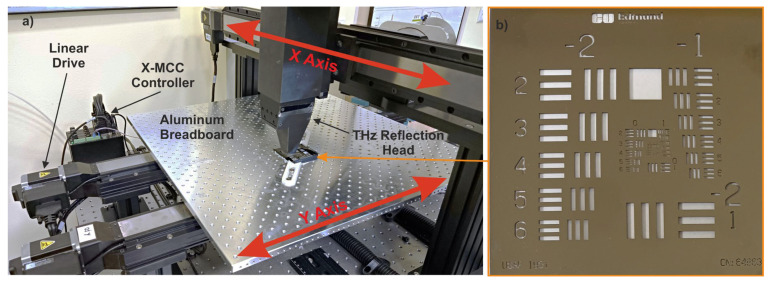
Photos of (**a**) an XY cartesian scanning gantry, and (**b**) a USAF 1951 test sample.

**Figure 4 sensors-24-01806-f004:**
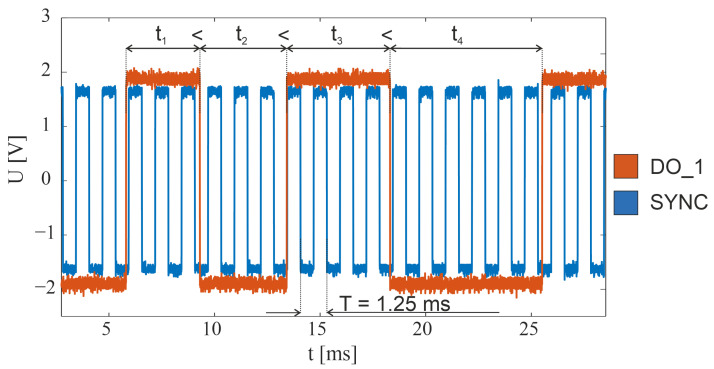
Oscillograms of synchronization signals in the THz scanner system.

**Figure 5 sensors-24-01806-f005:**
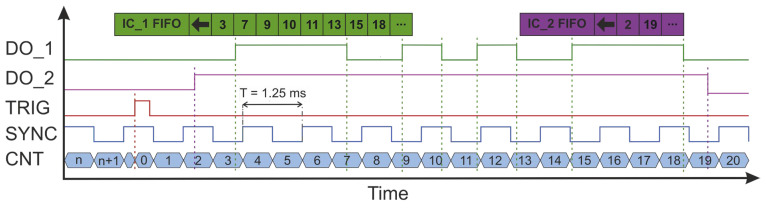
Oscillograms of synchronization signals in the THz scanner system. The edges of each signal mean the following: DO_1—drive shift by a given step, DO_2—start and end of drive movement, TRIG—THz data reading trigger, SYNC—THz pulse acquisition. The tables above the graph symbolize first input first output (FIFO) buffers that store subsequent counter values captured at edges of DO_1 (IC_1 FIFO) and DO_2 (IC_2 FIFO) signals.

**Figure 6 sensors-24-01806-f006:**
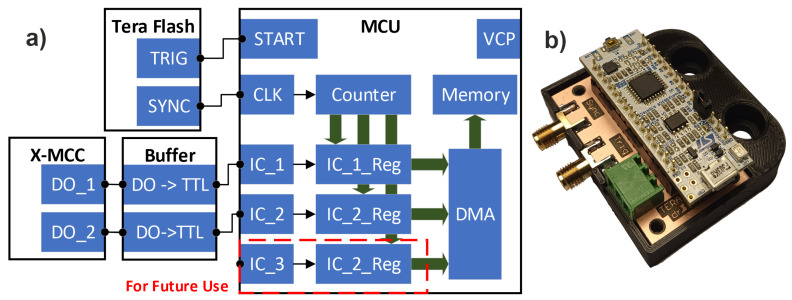
(**a**) Block diagram of the device implementing encoder-based synchronization; START—digital line used to trigger THz data read; CLK—counter clock input; IC_1, IC_2, and IC_3—counter capture inputs; IC_1_Reg, IC_2_Reg, and IC_3_Reg—counter capture registers; DMA—direct memory access controller; CVP—virtual com port; (**b**) a photo of the built system.

**Figure 7 sensors-24-01806-f007:**
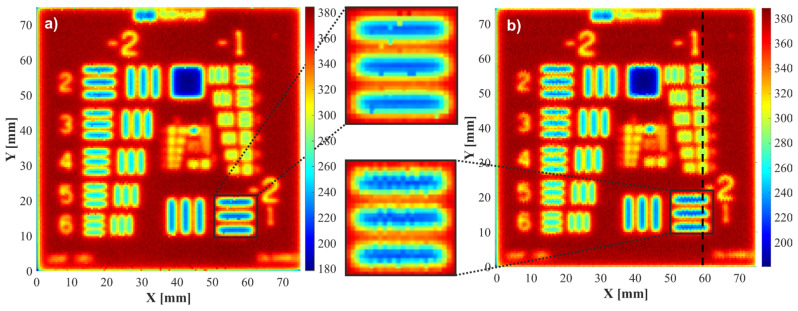
Experimental data: comparison of images obtained by scanning the USAF 1951 test using (**a**) the step-by-step method, and (**b**) the synchronized scan method at the fastest scanning speed possible.

**Figure 8 sensors-24-01806-f008:**
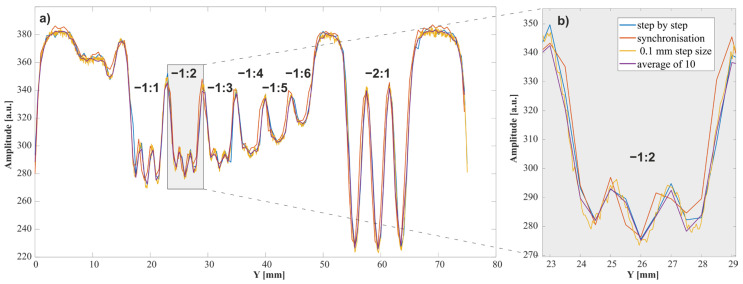
(**a**) Experimental data: vertical cross-section for the X = 59 mm line of the USAF 1951 test; (**b**) a fragment of a line containing element −2 from the −1 group. The values on the graph indicate the group number: element number in the USAF test.

**Figure 9 sensors-24-01806-f009:**
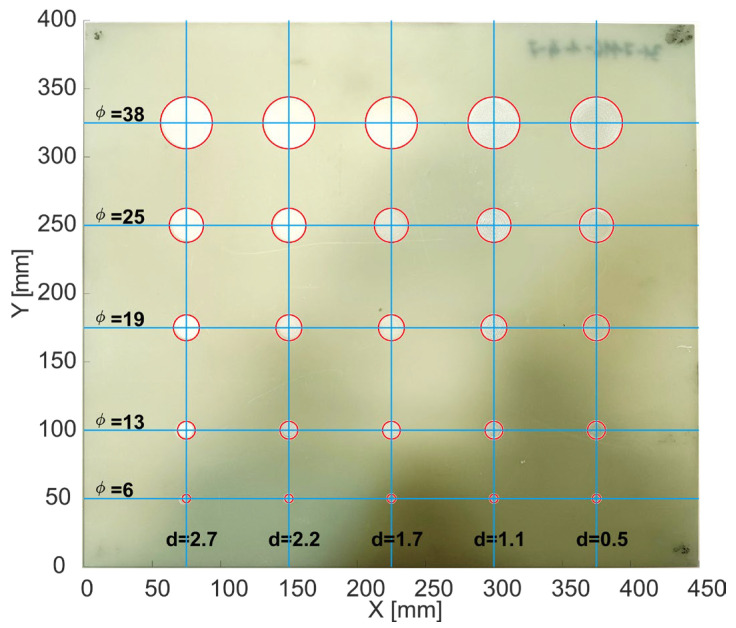
Photo of the sample with marked holes (red circles); *φ* is the diameter of the hole and *d* is the hole depth.

**Figure 10 sensors-24-01806-f010:**
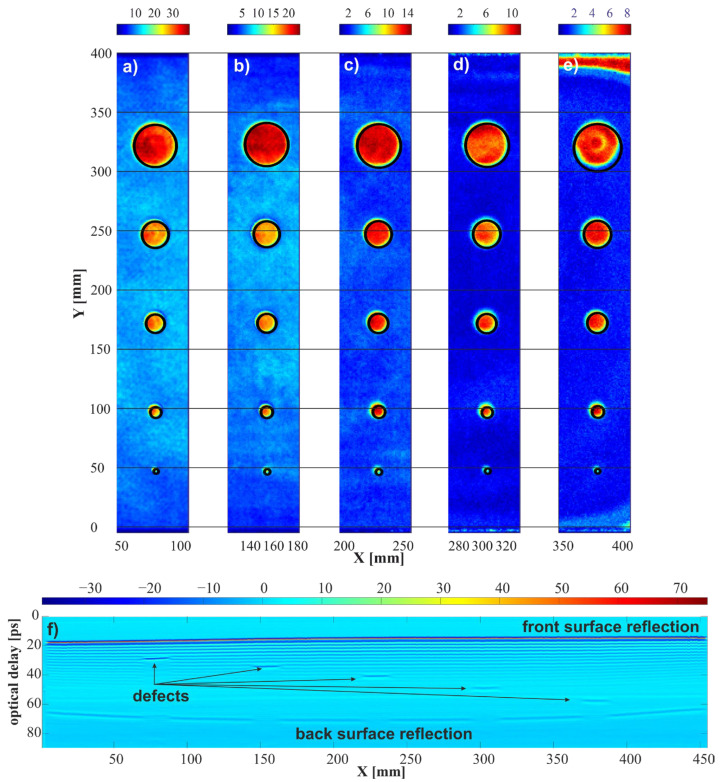
The results of the image analyses made with the developed synchronization method. C-scans determined for a fragment with a width of 4.5 ps starting at (**a**) 25 ps; (**b**) 31 ps; (**c**) 38 ps; (**d**) 47 ps; and (**e**) 56 ps. (**f**) Horizontal B-scan for Y line 172 mm.

**Table 1 sensors-24-01806-t001:** Parameters of the defects found in the image. All dimensions are in millimeters (mm).

Actual Defect Diameter	Defect Diameter in Image	*Y*-Axis Defect Position	Medium Distances between Defects
6	4.6 ± 0.4	46.7 ± 0.1	75.0 ± 0.3
13	9.8 ± 0.5	96.8 ± 0.2	75.0 ± 0.3
19	16.3 ± 0.5	171.7 ± 0.2	75.0 ± 0.4
25	22.5 ± 0.5	246.9 ± 0.2	75.0 ± 0.4
38	37.1 ± 1.7	321.5 ± 0.3	75.1 ± 0.7

## Data Availability

The raw data supporting the conclusions of this article will be made available by the authors on request.

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
