# Peer review of "A Hardware Encoder-Based Synchronization Method for a Fast Terahertz TDS Imaging System Based on the ECOPS Scheme"

_sensors, 2024, doi:10.3390/s24061806_

Round 1

Reviewer 1 Report

Comments and Suggestions for Authors

The authors have presented very nice work which contributes to the existing literature. However, there are some suggestions which I request in order to further improve the quality of the manuscript.

1) Clearly mention the experimental data and setups. 

2) How synchronization of hardware and software is achieved.

3) How accurate is the proposed synchronization techniques for TDS imaging? It would be better to have comparative study to support the validity of the proposed technique. 

4) please highlight the novel contribution and how these are different than the existing work

5) Literature survey can be improved.

Comments on the Quality of English Language

No major issues in the language detected except some typos. 

Reviewer 2 Report

Comments and Suggestions for Authors

Authors are reporting a proof of concept work. Even if a step vise raster scanning platform is a well-established method, the possible synchronization problems in the presence of a heavy objects sampled at a very large surface area is a technical problem reducing the acquisition speed. The authors are applying a box car type integration methodology  to achieve very high mapping speeds by overcoming  the synchronization problems in between the stage controllers and the THz detection platform in an effective and clever way. The work is well organized and the chosen concepts are  clearly represented. The introduction provides sufficient background and includes relevant references.

I would like to thank all the Authors for their efforts and I kindly ask them to address my comments and suggestions below:

1.  The text is clearly written and well presented. The English language and style are fine still an overall spell check would be beneficial.

2. I would suggest the authors to add a mapping sample acquired at the fastest scanning speed possible and comparing with a traditionally acquired image.

3. The introduction provides sufficient background yet, would be good to add some recent literature related to a large surface area THz mapping and motion control related works. I would suggest the authors to cite to:

Terahertz Cross-Correlation Spectroscopy and Imaging of Large-Area Graphene/ Sensors 2023, 23(6), 3297;

A Large Area Wide Bandwidth THz Phase Shifter Plate for High Intensity Field Applications / Photonics, 2023, 10(7), 825

THz Multi‐Mode Q‐Plate with a Fixed Rate of Change of the Optical Axis Using Form Birefringence / Micromachines, 2022, 13(5), 796

J-Net: Improved U-Net for Terahertz Image Super-Resolution/ Sensors 2024, 24(3), 932

High Precision Motion Compensation THz-ISAR Imaging Algorithm Based on KT and ME-MN /Remote Sens. 2023, 15(18), 4371;

The manuscript by Maciejewski et.al, is already a well organized and clearly represented work especially for the researchers working specifically in the field. To help readers from diverse backgrounds it would be beneficial to add further details on some of the specific points mentioned in the text.    I thank you for your attention and following your suggestions I have included a few more comments and suggestions to the authors, please find enclosed below. Please update it in the system accordingly.

4. Moreover, I would suggest the authors add  further comments comparing with other published materials related to Precision Motion Compensation and fast scan imaging routines.

5. the quality of the figures are satisfactory, yet must be improved in terms of some basic typos and missing information:

For instance in figure 3: “thz head”  THz Head

It could also be better to identify what the head is? In terms of the opt mechanical scheme utilized inside the system head?

In figure 4: (shift by a given step (DO_1) or end of line DO_2)., DO_2 must be better expressed in the figure.

Moreover, a further explanation regarding the box-car type detection methodology, how it is achieved, how the hardware solution works and also, in figure 5: Oscillograms of synchronization signals in the THz scanner system must be more clearly explained… What is FIFO is not clear. Further explanation must be added to help readers from diverse backgrounds.

5. Block diagram of the device implementing encoder-based synchronization must be further explained with details.

6. Figure 8. (a) Vertical cross-section for the X = 59 mm line of the USAF 1951 test; (b) a fragment of a 305 line containing element -2 from the -1 group. The indicated values -1:2 ratios are not clearly expressed…. A detailed explanation and discussion is needed.

7. a range from -5 to 460 mm in the X-axis, and from -5 to 410 320 mm in the Y-axis.

A negative range is not clear, since the system utilizes high precision controllers with micron steps, the error seems to be far above the specified control limits. This must be further clarified and better expressed.

8.  The authors show a defect diameter smaller than the actual size. In a general size due to the diffraction and or reflection at the edges, the image is expected to have a larger size due to the errors mentioned… this point must be better explained in the text.

9.  The authors claim “scanning time which is 30 times faster than that achieved with the tra-370 additional step-by-step method” must be clearly demonstrated with calculations. Further information on the scanning speed parameters choice must be added. 

10. the conclusion must be further improved explaining, What specific improvements should the authors consider regarding the methodology? What further controls should be considered?  What parts do you consider original or relevant for the field? What specific gap in the field does the paper address?

Comments on the Quality of English Language

The text is clearly written and well presented. The English language and style are fine still an overall spell check would be beneficial.

Round 2

Reviewer 1 Report

Comments and Suggestions for Authors

The authors addressed all of my concerns. As I don't have any further concerns, so it can be accepted in my opinion. 

Comments on the Quality of English Language

English is okay. Minor typos and grammatical errors can be checked. 

Reviewer 2 Report

Comments and Suggestions for Authors

I would like to thank all the Authors for their efforts,

The quality of the work and the presentation have been improved in the revised version. The authors have answered my comments satisfactorily for the parts of major importance.

I would recommend publication in its present form.

With my best regards,